# Simulating sea level extremes from synthetic low-pressure systems

Jani Särkkä[1], Jani Räihä[1], Mika Rantanen[1], and Matti Kämäräinen[1]

[1] Finnish Meteorological Institute, P.O.Box 503 , FI-00101 Helsinki, Finland

**Correspondence:** Jani Särkkä (Jani.Sarkka@fmi.fi)

**Abstract.** In this article we present a method for numerical simulations of extreme sea levels using synthetic low-pressure systems as atmospheric forcing. Our simulations can be considered as estimates of the high sea levels that may be reached when a low-pressure system of high intensity and optimal track passes the studied region. We test the method using sites located along the Baltic Sea coast and simulate synthetic cyclones with various tracks. To model the effects of the cyclone properties on sea level, we simulate internal Baltic Sea water level variations with a numerical two-dimensional hydrodynamic model, forced by an ensemble of time-dependent wind and air-pressure fields from synthetic cyclones. The storm surges caused by the synthetic cyclones come on top of the mean water level of the Baltic Sea, for which we used a fixed upper estimate of 100 cm. We find high extremes in the northern Bothnian Bay and in the eastern Gulf of Finland, where the sea level extreme due to the synthetic cyclone reach up to 3.5 meters. In the event that the mean water level of the Baltic Sea has a maximal value (1 meter) during the cyclone, highest sea levels of 4.5 meters could thus be reached. We find our method to be suitable for use in further studies of sea level extremes.

## 1 Introduction

A large part of the population and infrastructure in the Baltic Sea region is concentrated in the vicinity of the coastline. Therefore, it is crucial to consider the potential for extreme sea level conditions when ensuring the safety of citizens and implementing coastal flood protection measures. To quantify the coastal flooding risks, a comprehensive investigation of the present-day sea level variability and the possible scenarios for mean sea level in the future is essential (Pellikka et al., 2018; Leijala et al., 2018; Pellikka et al., 2023). As extremely high sea levels in the Baltic Sea are primarily caused by mid-latitude cyclones, the impact of atmospheric conditions on sea level extremes is an important research topic.

Sea level is measured by tide gauges situated on the coastline. For many locations on the Baltic Sea coast, tide gauge records are only available for the last 100-200 years, although historical floods can be reconstructed from other sources, e.g. historical records or studies of coastal sediments (Rutgersson et al., 2022). Nevertheless, it is possible that many sea level extremes of the last 1000 years have not left any traces in the historical record, especially in historically sparsely populated coastal areas such as around the coast of the Bothnian Bay. In addition, in the northern Baltic Sea, the coastline has retreated due to land uplift, and the coastal strip affected by flooding in the past is now further from the shore. Therefore, the effects of high sea levels have been reduced by the relative decrease in mean sea level with respect to the coastline.

The Baltic Sea is a shallow, semi-enclosed sea connected to the Atlantic Ocean through the narrow and shallow Danish Straits. Local short-term variations in sea level are caused by winds, air pressure, currents, tides and internal oscillations (seiches). Long-term changes are caused by global mean sea level variations, post-glacial land uplift and the water flow between the North Sea and the Baltic Sea through the Danish Straits (Leppäranta and Myrberg, 2009; Johansson et al., 2014). Sea level extremes in the Baltic Sea are highest during the winter months, most commonly in November, December and January (Wolski and Wiśniewski, 2023). These short-term sea level extremes (on hourly or daily timescales) are usually caused by moving extratropical cyclones which in favorable atmospheric conditions can strengthen to powerful windstorms (Rantanen et al., 2024). Like sea level extremes, extratropical cyclones are most intense during the winter months, with about five windstorms per month in northern Europe (Laurila et al., 2021).

The effects of short-term sea level fluctuations are largest in the narrow bays of the Baltic Sea. In particular, the narrow Gulf of Finland is vulnerable to coastal flooding due to windstorms, such as Storm Gudrun, which caused damage in 2005 (Tõnisson et al., 2008; Averkiev and Klevannyy, 2010). Major coastal flooding occurred in St. Petersburg in 1777, 1824, and 1924 (Kulikov and Medvedev, 2017), of which the 1824 flood was the most extreme, with a maximum sea level of 4.21 m. Even in cases where in situ sea level measurements are not available, traces of extreme sea level events can be found in sediment studies (Piotrowski et al., 2017; Leszczyńska et al., 2022). More information on sea level extremes in the Baltic Sea can be found in recent reviews considering the region (Weisse et al., 2021; Rutgersson et al., 2022).

The probabilities of extreme sea levels are usually estimated applying extreme value analysis methods to observed sea level data (Arns et al., 2013). Extreme weather systems, such as tropical cyclones, rarely occur multiple times in a single location, so the extreme sea levels they cause are either unobserved or represented by single outliers. This makes it difficult to analyse extreme values of sea level data, especially in areas affected by tropical cyclones (O'Grady et al., 2022). However, even in the Baltic Sea, extratropical cyclones can cause extreme outliers in the statistics due to the shallow depth and shape of the coastline. For example, the highest sea level in the Gulf of Riga caused by Storm Gudrun in 2005 was an extreme outlier in the distribution (Mäll et al., 2017). Therefore, estimates of extremes based on statistical methods using observational data may be too conservative if cyclones caused by the highest sea level have not occurred at the location studied during the measurement period.

To overcome this limitation, studies have developed idealised low-pressure systems that can be used to investigate how high water levels could be caused by cyclones with optimal tracks and intensities in the Gulf of Finland (Averkiev and Klevannyy, 2010; Kalyuzhnaya et al., 2015; Apukhtin et al., 2017) and the Bothnian Bay (Gordeeva and Klevannyy, 2020). In a recent study (Andrée et al., 2022), synthetic uniform wind fields were used in extreme sea level simulations and an empirical-statistical method was used to model the site-specific relationships between wind and extreme sea level. In addition, synthetic cyclones were used to assess the hazards of high sea level and waves in tropical regions (Leijnse et al., 2022; O'Grady et al., 2022).

In this study, we aim to estimate the highest plausible limits of extreme sea levels in the Baltic Sea caused by strong extratropical cyclones. To investigate how high sea levels a storm with an optimal track could cause along the Baltic coast, we reconstruct idealised cyclones that are physically realistic but have not occurred during the period of tide gauge observations.

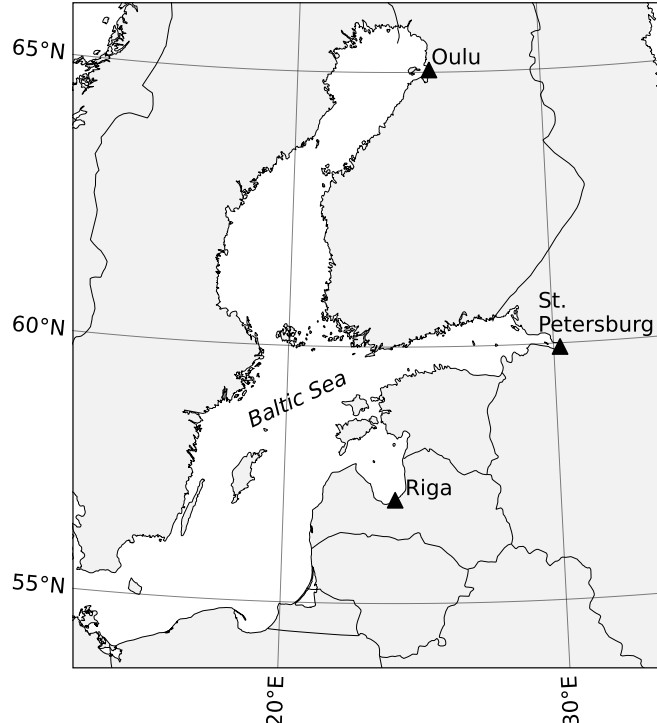

**Figure 1.** The Baltic Sea and locations (Oulu, St. Petersburg and Riga) where high sea levels were studied.

We describe the cyclones by a pressure field with a spatially varying time-dependent function that reproduces typical cyclone characteristics. The pressure and wind data are used as input to a numerical sea level model to simulate extreme sea levels.

## 2  Data and methods

### 2.1  Synthetic atmospheric forcing

To model the sea levels associated with our synthetic cyclones, we need information on the atmospheric conditions for the
forcing of the sea level model. To this end, we use a method where the mean sea level pressure and surface winds of the cyclone are calculated from the pressure field of a cyclone propagating with a constant velocity. The pressure field has the form of a Gaussian function

$$P_{x,y} = P_0 + \Delta P \exp\{-\frac{(x-x_0)^2}{2\sigma^2} - \frac{(y-y_0)^2}{2\sigma^2}\}, \tag{1}$$

where $P_0$ is the air pressure sufficiently far from the cyclone center, $\Delta P$ is the pressure anomaly in the cyclone center, $x_0$
and $y_0$ are the time-dependent coordinates of the cyclone center, and $\sigma$, the width of the pressure anomaly, is related to the

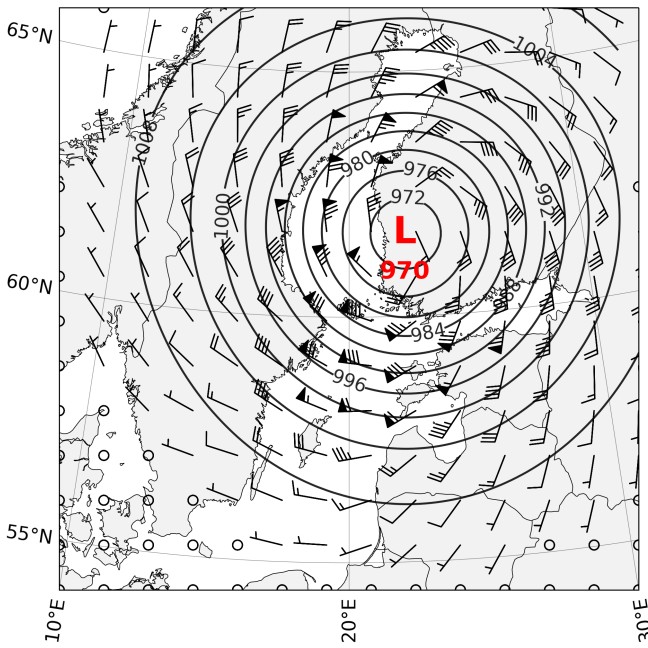

**Figure 2.** Example of synthetic cyclone and induced winds. Contours represent sea level pressure (hPa) and wind barbs represent 10 m winds (knots).

radius $R$ of the cyclone via $\sigma = 0.25R$. 10-metre geostrophic wind components $(u_g, v_g)$ are calculated from the pressure field using the geostrophic balance as

$$u_g = -\frac{1}{f\rho}\frac{\partial P}{\partial y}, v_g = \frac{1}{f\rho}\frac{\partial P}{\partial x}, \tag{2}$$

where $u_g$ (m s$^{-1}$) is the zonal geostrophic wind component, $v_g$ (m s$^{-1}$) is the meridional geostrophic wind component,
$f = 2\Omega\sin(\phi)$ is the Coriolis parameter ( $\Omega$ is the rotation rate of the Earth and $\phi$ is the latitude) and $\rho$ is the density of air (1.3 kg m$^{-3}$). An example of the pressure and wind field of a synthetic cyclone is shown in Fig. 2.

Actual wind components $(u, v)$ differ from the geostrophic winds $(u_g, v_g)$, especially near the surface where friction acts to slow winds down, leading to winds turning anticlockwise in the Northern Hemisphere, and where curvature effects (the centrifugal force) are important. We study this difference between the theoretical geostrophic and the actual near-surface
wind by using the analysis previously applied by Särkkä et al. (2017), with linear regression applied separately for wind speed and direction in the ERA-Interim reanalysis dataset (Dee et al., 2011) over the period 1979–2012. We obtain spatially varying coefficients for wind speed and direction change by comparing the actual 10-metre winds of the reanalysis dataset with geostrophic winds that we calculate from the mean sea level pressure field. We then modify the geostrophic winds in the cyclone generator with coefficients from Särkkä et al. (2017) to achieve more realistic wind fields. Finally, we save these wind
fields and pressure fields in a numerical grid and use them as the input of the sea level model.

## 2.2 Limitations of the synthetic cyclone

It should be noted that the use of synthetic cyclones is inevitably subject to certain limitations. One of these is the Gaussian shape of the cyclone. As can be seen from Fig. 2, the spatial field of the simulated cyclone is not entirely realistic, as the cyclone is symmetrical and lacks the frontal structures typical of extratropical cyclones. However, the Gaussian shape was originally chosen for two main reasons: (1) it requires only two parameters to describe it, and (2) it describes the general shape of cyclones with sufficient accuracy. It is clear, of course, that not all possible variations in cyclone shape can be described by the Gaussian simplification. However, we would like to emphasise the importance of other factors in determining sea level variations. For example, far more relevant to simulated sea levels than the deviations of the shapes from Gaussian are (1) the depth and extent of the cyclone, and (2) the track and propagation speed of the cyclone.

Furthermore, the pressure anomaly of the simulated cyclone is not time-dependent, neglecting the deepening and filling of the cyclone. The value of the pressure anomaly affects the simulated sea level mainly when the cyclone moves over the sea. In the Baltic Sea, cyclone passes the sea area in less than 24 hours, limiting the effect of the time-varying pressure anomaly on the simulated coastal sea level.

## 2.3 Cyclone parameters

The horizontal extent and depth of the synthetic cyclone, related to the radius $R$ and $\Delta P$, must be limited by physical considerations. According to Rudeva and Gulev (2007), the effective radius of cyclones over the ocean can be greater than 900 km, with an average of about 700 km. Therefore, we set the radius $R$ to 1000 km, although we acknowledge that this value may be at the upper limit of what is plausible in the Baltic Sea region (Rudeva and Gulev, 2007).

The pressure anomaly $\Delta P$ was constrained by the resulting geostrophic wind speeds. Based on observations and previous work, we decided that surface mean wind speeds should not exceed 40 m/s. This limit roughly corresponds to the maximum wind speeds observed during the Storm Gudrun (mean wind speed 33 m/s, wind gust 42 m/s, Valinger and Fridman, 2011), and 40 m/s was also the highest stationary wind speed examined in a recent study of wind-driven extreme sea levels (Andrée et al., 2022). With this constraint, the pressure anomaly $\Delta P$ was set to -40 hPa.

The numerical sea level model describes how surface winds cause water flow through the wind friction component in the Navier-Stokes equations. The dependence of the wind friction on velocity is quadratic in our model, but in reality this dependence is likely different for extreme wind speeds (over 50 m s$^{-1}$). This is an additional argument for limiting the surface wind speeds to 40 m s$^{-1}$ for the synthetic cyclones. We fix the radius of the cyclone and the pressure anomaly using the values mentioned above (which limits also wind speeds), and vary four parameters in the simulations: the latitude and longitude of cyclone origin, plus the speed and direction of cyclone propagation (cyclone velocity and direction given by zonal speed $u$ and meridional speed $v$). We estimate the distribution of mean cyclone velocities for storms that pass the Baltic Sea region during October-March 1979-2020 based on ERA5 reanalysis data (Hersbach et al., 2020) using the same cyclone track dataset as was used in Laurila et al. (2021). The smallest velocities from this analysis are around 5 m s$^{-1}$ (Fig. 3a). Even smaller velocities

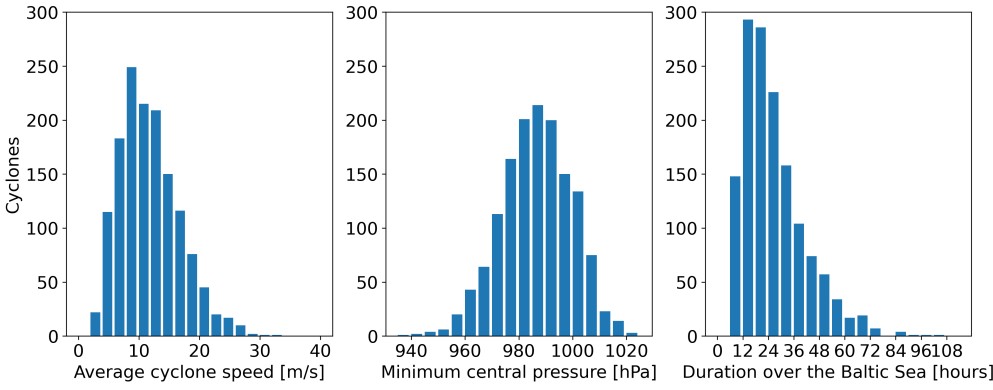

**Figure 3.** Distribution of mean speeds, minimum pressures and durations of cyclones that pass the Baltic Sea region in October-March 1979-2020 in ERA5 reanalysis.

of the cyclone lead to situations where the cyclone is almost stationary, with cyclone winds piling water towards the coast. As this scenario is likely unrealistic, we limit the minimum velocity of the cyclone to 5 m s$^{-1}$.

## 2.4 Numerical sea level simulation

In our numerical simulations we use a barotropic sea level model which describes the internal (intra-basin) fluctuations of the Baltic Sea with a set of two-dimensional hydrodynamic equations (Häkkinen, 1980). These equations are based on shallow water equations, derived from the Navier-Stokes equations by integrating the vertical coordinate out. Surface winds and surface pressure are used as atmospheric forcing for the sea level model. Computational domain extends from 53 N to 66 N and from 9 E to 31 E. The spacing of the numerical grid is 0.1 degrees in the meridional direction and 0.2 degrees in the zonal direction. We treat the Baltic Sea as a closed basin, with no connection through the Danish Straits to the North Sea. The model has previously been validated with Finnish tide gauge data and with ERA-Interim forcing 1979-2012 (Särkkä et al., 2017). The correlation between hourly data of simulated and observed sea levels was from 0.91 to 0.94 for all 13 Finnish tide gauge sites, even if the spacing of the numerical grid was sparser (0.25 X 0.5 degrees) than in this study. The details of the sea level model were explained in Särkkä et al. (2017).

The variations in the water volume of the Baltic Sea are mainly caused by the flow of the water in and out of the Baltic Sea through the Danish Straits. The contribution of the water volume to the local sea level (also called preconditioning) is termed the water balance component. This component can be estimated with a statistical model, based on wind speeds at a single coordinate point at (55° N, 15° E) (Johansson et al., 2014; Särkkä et al., 2017). The progression of a cyclone over the Baltic Sea region in the high sea level cases takes only one or two days, and the change in the water balance within this time is small (less than 10 cm). Thus we do not include changes in the water balance in our simulations. Instead, we estimate the extreme value of the water balance component to be 100 cm (Särkkä et al., 2017) and add it to the simulated local sea level maximum.

This describes the situation when preceding weather conditions have raised the water volume in the Baltic Sea, and then the arriving cyclone causes fluctuations on top of the high mean sea level.

The tides are small in the Baltic Sea, with largest tidal ranges (over 20 cm) at the eastern end of the Gulf of Finland and near the Danish Straits in the southwestern Baltic Sea. For a large part of the Baltic coast, the tidal range is only a few cm (Medvedev et al., 2016), and thus we do not include tides in the model.

## 2.5   Simulation parameters

As we are interested in the highest sea levels on the Baltic Sea coast, we select three sites to represent different bay areas of
the Baltic Sea, including coastal cities where the sea level variability is large and where the effects of flooding are significant - the latter due to dense inhabitation with substantial infrastructure on low-lying coastlines. As the predominant direction of the cyclone tracks is from the west, we choose to study sites at the ends of bays, on the eastern coast of the Baltic Sea. At these locations, the effect of wind-driven water transport is large, as the water piles towards the coastline. We select the locations so that the Gulf of Bothnia is represented by Oulu, situated at its northern end (Fig. 1). The Gulf of Finland is represented by St.
Petersburg, situated at its eastern end. The smaller sub-basin, the Bay of Riga is represented by Riga. In the Gulf of Bothnia, we also simulated sea levels at Kemi, but found that the highest simulated sea levels at Oulu exceeded the ones found at Kemi.

We use the surface wind and pressure fields of synthetic cyclones as atmospheric forcing for the sea level model. We compare simulation results with the observed extremes in tide gauge data from Fig. 1 of Averkiev and Klevannyy (2010). As the initial condition for the Baltic Sea level in the simulation is a uniform constant sea level, our results represent the sea level fluctuation
with respect to the prevailing mean sea level in the Baltic Sea preceding the arrival of the cyclone.

To compare the simulated extremes for different locations with the observed ones, we add 100 cm to the simulated extreme, which takes into account the high water volume in the Baltic Sea that often prevails before the extreme sea level situation. As our aim is only to estimate the local extremes to an accuracy of tens of centimeters, we use a fixed value of 100 cm for the initial sea level for all simulated sites with respect to the reference level, even if this level differs between the national vertical
reference systems used for different sites.

In Table 1 we give the parameters that yield the highest sea levels at each of the three sites. The origins of the cyclones are all on the longitude 10 E. The effect of the winds on the sea level fluctuation with the distance close to the radius from the cyclone center is small, thus is was found that it is not necessary to set the cyclone origin further in the west. A common factor for the cyclone tracks of these maxima was that the propagation speeds were close to the minimum speed limit $5.0 \text{ m s}^{-1}$ :
$5.7 \text{ m s}^{-1}$ for Oulu, $5.4 \text{ m s}^{-1}$ for St. Petersburg and $5.0 \text{ m s}^{-1}$ for Riga. Due to the low propagation speed, high wind speeds prevail for a longer time over the Baltic Sea, enhancing the water flow towards the ends of bays.

| Location | Origin | zonal vel. | meridional vel. | total velocity (m s$^{-1}$) | Max. sea level (cm) |
|----------|--------|-----------|-----------------|-----------------------------|---------------------|
| Oulu | (61.0, 10.0) | 4.3 | 3.7 | 5.7 | 294 |
| St. Petersburg | (57.5, 10.0) | 4.8 | 2.5 | 5.4 | 346 |
| Riga | (67.0, 10.0) | 3.3 | -3.7 | 5.0 | 265 |

**Table 1.** Cyclone parameters for the highest simulated sea levels at Oulu, St. Petersburg and Riga.

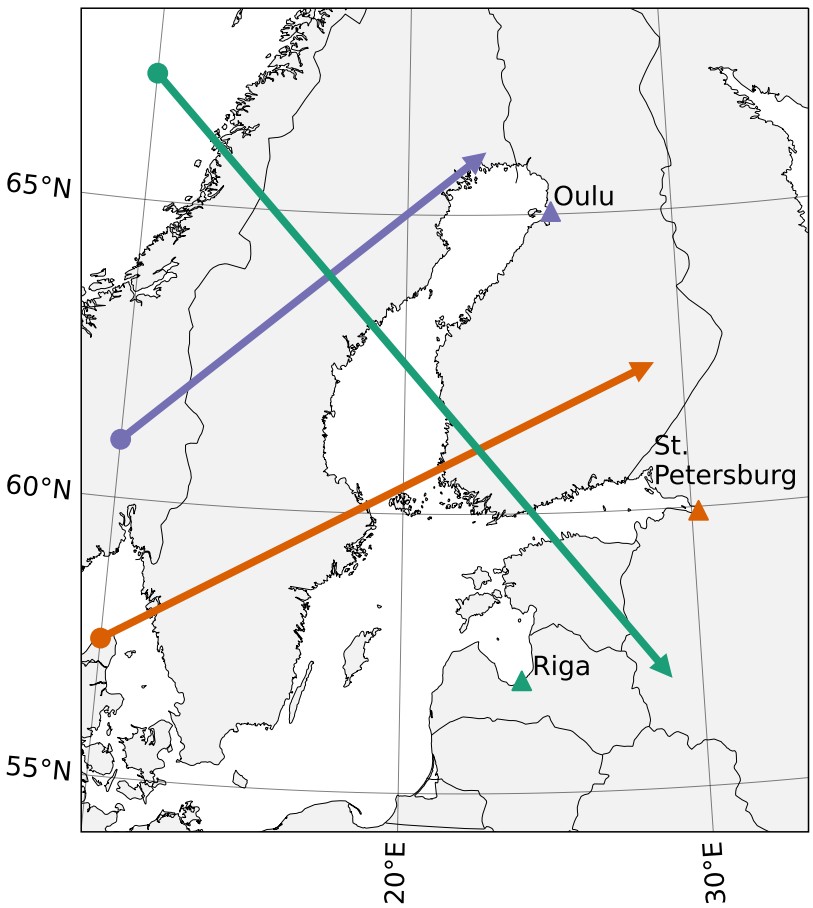

**Figure 4.** Cyclone tracks of the highest simulated sea levels at Oulu, St.Petersburg and Riga.

## 3   Results

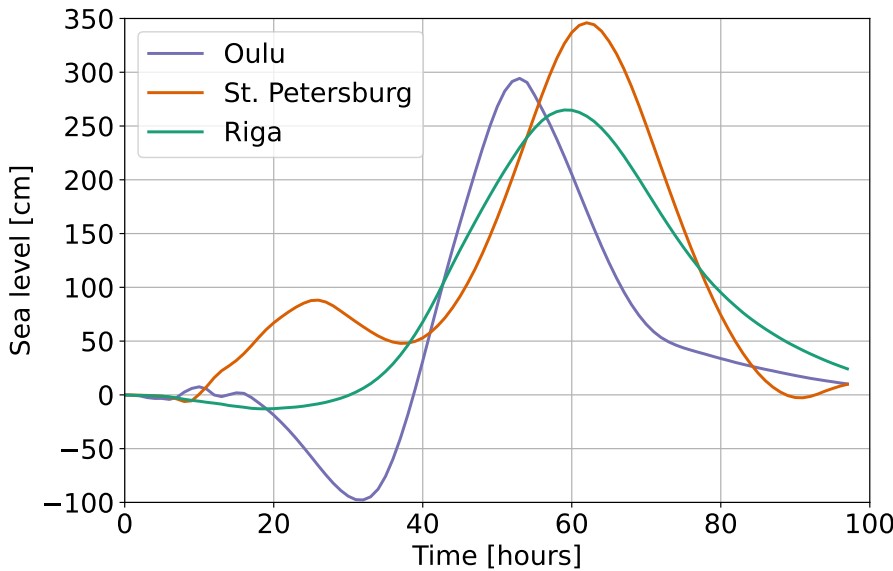

**Figure 5.** Highest simulated sea levels at Oulu, St.Petersburg and Riga.

The simulated sea level extreme at **Oulu** is caused by a cyclone progressing northeastwards from southern Norway towards Finnish Lapland (Fig. 4). The strong winds on the right hand side of the propagation direction generate water flow from the

170 central Baltic Sea towards the northern end of the Bothnian Bay. At first, the wind direction in front of the cyclone center is towards the northwest in the Bothnian Bay, moving water away from Oulu and causing a decrease in sea level down to -100 cm. Then, as the cyclone center moves northwest of the Bothnian Bay, the wind direction is towards the northeast, piling water towards the northeastern end of the bay, which causes a rapid increase in sea level at Oulu. Finally, when the cyclone is north of the Bothnian Bay, the wind direction is from west to east over the northern Bothnian Bay. This pushes yet more water towards

the Oulu coastal region, leading to a very high sea level maximum of 294 cm (Fig. 5). The highest observed sea level at Oulu is 183 cm from January 14 1984. This event was caused by a deep low pressure (950 hPa) having its center in Sweden, inducing southern winds that both increased the water amount in the Baltic Sea and pushed water towards the end of the Bothnian Bay. Taking into account the maximum water balance component (100 cm), a sea level at Oulu of 394 cm could be reached, which is significantly higher than the observed maximum value. In the study of Andrée et al. (2022) with uniform wind fields (wind

speed 22 m s$^{-1}$), the highest simulated sea levels in the Bothnian Bay were at Kalix and Kemi (both situated west of Oulu), where extreme sea levels between 275 and 300 cm were obtained. These levels are of the same magnitude as our result for the extreme sea level at Oulu without the water balance component.

Räty et al. (2023) used Bayesian hierarchical modeling to analyse return level estimates and theoretical upper limits for the sea level on the Finnish coast from tide gauge data. At Oulu, the 1000-year return level is approximately 200 cm and the

theoretical upper limit for the sea level using Bayesian modeling is approximately 250 cm. These estimates include the water balance component, and are considerably lower than the nearly 4 meters estimate in our analysis. This indicates that the high values that we obtain from our synthetic cyclones method can not always be derived from the statistical extreme value analysis.

The cyclone track that causes the highest simulated sea level extreme at **St. Petersburg** (346 cm, Fig. 4) is interesting. The cyclone enters the Baltic Sea region from northern Denmark and moves over southern Finland. Winds cause water to flow from the central Baltic Sea to the Gulf of Finland, which accumulates water at its eastern end. The highest observed sea level occurred in 1824, reaching 421 cm at St. Petersburg. In the study of Averkiev and Klevannyy (2010), the pressure anomaly was a time-dependent exponential function, having maximum pressure anomaly -50 hPa (minimum pressure 960 hPa at (61.7 N, 29.0 E)). The maximum wind speed in the Gulf of Finland was 41 m/s, and the velocity of the cyclone was 14.24 m/s. The initial position of the cyclone center was at (60 N, 10 E), and the cyclone track crossed 30 E at the Finnish-Russian border. In our simulation (Fig. 4), the initial position is more southern (57.5, N 10 E), but the crossing of 30 E is north of the track of the cyclone in Averkiev and Klevannyy (2010), and the velocity is slower (5.4 m/s). The simulated sea level at St. Petersburg by Averkiev and Klevannyy (2010) was 590 cm, where the initial mean sea level was set to 0 cm. The rapid deepening of the pressure anomaly over the Gulf of Finland leads to this extreme value in Averkiev and Klevannyy (2010), whereas in our simulation the piling up of water in the end of the Gulf of Finland is the main factor in the extreme sea level 346 cm. Taking into account the maximum water volume of the Baltic Sea by adding 100 cm, the highest estimated sea level in our simulations at St. Petersburg is 446 cm. The narrow shape of the Gulf of Finland and its west-east orientation leave the eastern end of the Gulf of Finland especially vulnerable to sea flooding when weather conditions induce water transport towards St. Petersburg.

The highest simulated sea level maximum at **Riga** is due to a cyclone approaching from in central Norway, then moving to the southeast over the Bothnian Sea and southwestern Finland (Fig. 4). Northerly winds push water from the Bothnian Sea and central Baltic Sea towards the Bay of Riga. As the cyclone moves over the Gulf of Finland and Estonia to Russia, the northwesterly winds cause water accumulation in the vicinity of Riga. The simulated maximum at Riga is 265 cm. The highest observed sea level is 229 cm from November 1969. By adding the water balance component of 100 cm, the highest estimated sea level is 365 cm, considerably higher than the observed maximum.

In summary, the simulated extremes clearly exceed the observed ones at Oulu and Riga, even accounting for a tens of centimeters ambiguity in the value of the water balance component from the reference level of the tide gauge location. At St. Petersburg, however, the simulated extreme is of similar height to the observed extreme in 1824.

## 4  Conclusions

The results of our study show that a numerical sea level model combined with synthetic cyclones can be used to study the characteristics of those cyclones that induce extreme sea levels on the Baltic coast. Comparing the three studied sites, we find the highest simulated sea levels at St. Petersburg, - around 3.5 meters, - so that, adding the water balance component, sea levels over 4 meters are possible. These values are comparable with the values observed during the extreme flood of 1824 (421 cm

in St. Petersburg). As the flood in 1824 was the highest in the 300-year history of St. Petersburg, the simulated case could correspond to the extreme weather conditions that contributed to the sea levels observed in 1824.

The high sea level at Oulu, approximately 4.0 meters, is due to a cyclone causing water flow from the entire Gulf of Bothnia towards the northern Bothnian Bay, and finally the change in wind direction from westwards to eastwards in the Bothnian Bay. One could expect that an even higher sea level would be found in Kemi, situated at the northern end of the Bothnian Bay, as the highest sea level in Finland (201 cm in 1982) was observed there. In our simulations at Kemi (results not discussed in this publication), the highest maxima are all under 4.0 meters. The shape of the coastline and bathymetry in the Bothnian Bay apparently favor high sea levels at Oulu more than at Kemi. Our simulation results indicate that there is potential for a flood exceeding four meters at Oulu, far surpassing the observed maximum in 1984 (183 cm).

At Riga, the highest simulated sea level is obtained with a cyclone passing the Baltic Sea with a track from central Sweden towards northwestern Russia. The simulated extreme of 265 cm (with 1 m added to include the water balance component) is considerably higher than the observed maximum (229 cm) in 1969. The cyclone related to the maximum in 1969 had a similar track compared to the one used in our simulation. This indicates that a similar storm track can occur and that a cyclone with such a track and slow velocity could cause high sea levels in the Riga region.

The effect of waves on the extreme sea levels on the coast was not considered in this study, as this effect strongly depends on the bathymetry and on the location. Our aim was not to evaluate rigorously the highest sea level taking into account all factors, but to evaluate what is the contribution of the synthetic cyclone in the sea level maximum, which already exceeds observed sea level extremes in many locations on the Baltic Sea coast. For detailed studies of extreme sea levels for a single site, simultaneous sea level and wave simulations are needed, and such studies have already been made in the Baltic Sea (Apukhtin et al., 2017; Gordeeva and Klevannyy, 2020).

Our results indicate that synthetic cyclones are a useful tool for studying the sea level response to cyclones with different characteristics. In particular, our method offers the opportunity to study those extreme sea levels which can be expected on the Baltic Sea coast if all the components that raise the water level are present, namely a high water balance paired with a slow and deep cyclone moving along an optimal track. Our simulations of extreme sea levels reveal those areas of the Baltic Sea which, due to shoreline geometry and coastal bathymetry, are especially prone to extreme coastal flooding.

*Author contributions.* All authors contributed to the design of the study. JR and MK implemented the method for the generation of synthetic cyclones. JS implemented the numerical sea level model. MR studied the properties of cyclones from reanalysis data. JS performed the sea level simulations with synthetic cyclones and conducted the analysis presented in the paper, and wrote the manuscript with critical comments from the other authors.

*Competing interests.* The authors declare that they have no conflict of interest.

*Acknowledgements.* This research was funded by the Finnish State Nuclear Waste Management Fund (VYR) through the Finnish Research Programme on Nuclear Power Plant Safety 2019-2022 (SAFIR2022).

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
