# Peer review of "Simulating sea level extremes from synthetic low-pressure systems"

_Natural Hazards and Earth System Sciences, 2023_

## Referee Comment (RC2)

**Review**

On the manuscript **"Simulating sea level extremes from synthetic low-pressure systems"** by Jani Särkkä, Jani Räihä, Mika Rantanen, and Matti Kämäräinen.

**Overview: Conditional Acceptance Upon Minor Revision**

The paper presents simulated conditions of both Baltic sea level hydrodynamics and cyclone pressure conditions in an attempt to better understand extreme sea level variations caused by the latter. The employed method is sound and leads to interesting results. However, the manuscript lacks a proper review of the literature on the methods of extreme value theory on sea level or storm surge extremes as well as in the quantification of these extremes based on the aforementioned theory. Specific comments include:

1. The authors must remind the reader that there already exists a theoretical framework to estimate how much the extremes will deviate from the maximum of a typical Gaussian population. Following the seminal work of Gumbel (1958), the extreme value theory has found important applications on both global and local sea extremes. In global terms, I refer to studies such as of Butler et al. (2007), Arns et al. (2013) and O'Grady et al. (2022).

2A. While the authors correctly point out to the relevance of metocean parameters for the local effect of sea level change, such as wind and currents. To clarify the local effects of wind/waves/currents, the authors should mention that as waves become more nonlinear towards the shore they can decrease the sea level in about 5% of the significant wave height and through wave dissipation increase the sea level in up to 20% of the significant wave height (Bowen et al., 1968; Massel and Gourlay, 2000). For the effect of currents on the mean sea level, the authors could cite at least the theoretical work of Brevik (1978). Additionally, attention has to be made to the fact that extreme waves can further increase this oscillation in mean water level, and estimates from extreme value theory indicate that extreme heights can be increased by 10-30% depending on the sea conditions such as shoaling effects (Benetazzo et al., 2015; Barbariol et al., 2015, 2019; Bolles et al., 2019; Mendes and Scotti, 2020; Trulsen et al., 2020; Mendes et al., 2021), which can further amplify oscillations in the mean water level.

2B. In particular, given a distribution of a time series (let it be the mean sea level for example), one can compute the expected maximum extreme value following section 4b of Benetazzo et al. (2015) or section 3 of Mendes and Scotti (2020). I encourage the authors to attempt to compute this expected maximum of a Gaussian sea for the sea level and compare with their simulations. The authors should discuss the magnitude of the local wind-wave effects with that of the purely atmospheric pressure. At the very least, I expect the authors to discuss this alternative method.

3. There should be a review of mathematical modelling on cyclone pressure fields, and a discussion of why the particular choice (eq. 1) has been chosen.

4. The authors provide an ERA5 analysis of cyclone speeds that pass through the Baltic sea, but this information is not sufficient. They also need to display the average spatial (and vector) shape of these cyclones, not to mention the duration of their path in the Baltic Sea. Furthermore, a brief discussion (or figure display) of the intensity of the cyclone as they enter the Baltic Sea.

5. In section 2.3 the governing equations should be written down, and assumptions and limitations discussed thereof.

6A. The main results are presented in figures 5-6. The text should be clear on whether this analysis can be made only at a few locations, or if these locations were picked for a particular physical reason. Otherwise, I encourage the authors to provide a Baltic Sea analysis (as claimed in the text) instead of a few locations.

6B. While figure 6 is clear on the extremes, it should be normalized by the expected significant wave height as the cyclone passes. This comparison provides a better scale of the cyclone effects as compared to local wind/wave effects on sea level change.

6C. I encourage the authors to provide several contour plot panels showing the scale of normalized sea level change (by the significant wave height) across the entire Baltic sea coast. Each panel would show the sea level change at a particular time since the appearance of the cyclone.

6D. Pages 135-145 describe the path of the cyclone affecting the three cities of figure 5. Why not plot the path on a figure? It would better suit the manuscript and help the reader.

**Conclusion**

The reviewer thanks for the opportunity to read this important work. Overall, I support the publication of this preprint once all these minor issues have been clarified/amended.

**References**

Arns, A., Wahl, T., Haigh, I., Jensen, J., Pattiaratchi, C., 2013. Estimating extreme water level probabilities: A comparison of the direct methods and recommendations for best practise. Coastal Engineering 81, 51–66.

Barbariol, F., Benetazzo, A., Carniel, S., Sclavo, M., 2015. Space-time wave extremes: The role of metocean forcings. J. Phys. Oceanogr. 45, 1897–1916.

Barbariol, F., Bidlot, J.R., Cavaleri, L., Sclavo, M., Thomson, J., Benetazzo, A., 2019. Maximum wave heights from global model reanalysis. Progress in oceanography 175, 139–160.

Benetazzo, A., Barbariol, F., Bergamasco, F., Torsello, A., Carniel, S., Sclavo, M., 2015. Observation of extreme sea waves in a space-time ensemble. J. Phys. Oceanogr. 45, 2261–2275.

Bolles, C., Speer, K., Moore, M., 2019. Anomalous wave statistics induced by abrupt depth change. Physical Review Fluids 4.

Bowen, A., Inman, D., Simmons, V., 1968. Wave 'set-down'and set-up. Journal of Geophysical Research 73, 2569–2577.

Brevik, I., 1978. Remarks on set-down for wave groups and wave-current systems. Coastal Engineering 2, 313–326.

Butler, A., Heffernan, J.E., Tawn, J.A., Flather, R.A., Horsburgh, K.J., 2007. Extreme value analysis of decadal variations in storm surge elevations. Journal of Marine Systems 67, 189–200.

Gumbel, E. J., 1958. Statistics of Extremes. New York.

Massel, S.R., Gourlay, M.R., 2000. On the modelling of wave breaking and set-up on coral reefs. Coastal engineering 39, 1–27.

Mendes, S., Scotti, A., 2020. Rogue wave statistics in (2+1) gaussian seas i: Narrow-banded distribution. Appl. Ocean Res. 99, 102043.

Mendes, S., Scotti, A., Stansell, P., 2021. On the physical constraints for the exceeding probability of deep water rogue waves. Appl. Ocean Res. 108, 102402.

O'Grady, J., Stephenson, A., McInnes, K., 2022. Gauging mixed climate extreme value distributions in tropical cyclone regions. Scientific Reports 12, 4626.

Trulsen, K., Raustøl, A., Jorde, S., Rye, L., 2020. Extreme wave statistics of long-crested irregular waves over a shoal. J. Fluid Mech. 882.

---

## Author Comment (AC1)

We thank the reviewer for constructive comments of our submitted manuscript. The point-by-point replies to the comments of the reviewer are below. Your comments are marked in black and our responses in red.

Review on "Simulating sea level extremes from synthetic low-pressure systems" by Jani Särkkä, Jani Räihä, Mika Rantanen, and Matti Kämäräinen

The manuscript investigates sea level extremes in the Baltic Sea through numerical simulations of synthetic low-pressure systems. The authors conducted simulations based on historical records and compared the results with actual data. However, there are several areas where the manuscript can be improved before it is considered for publication:

**Cyclone Model Validation**

The article heavily relies on synthetic cyclones, which are artificially created. This approach introduces several assumptions, and synthetic cyclones may not accurately represent the complex dynamics and characteristics of real cyclones. This limitation raises questions about the reliability of the results. The discussion on the model's validity should be expanded in Section 2 instead of conclusion. To enhance the manuscript, the authors can:

- Justify the chosen intensity of the cyclone (pressure anomaly ΔP of -40 hPa and minimum radius R of 1000 km) by comparing it to extreme extratropical storm events observed in the Baltic Sea. What is the implication of the radius R and how is it different from the radius of the maximum wind of a tropical storm?
- Explain the characteristics of extratropical storms, compare with the model and discuss the limitation of the model
- If possible, discuss their similarities and differences compared to tropical storms and compare the synthetic cyclone model to existing hurricane models (e.g., Holland 1980).

Thank you for this comment. We agree that there are several assumptions involved in the design of synthetic cyclones. This is an inevitable issue whenever research is conducted with synthetic or artificially created cyclones.

The Gaussian shape was originally chosen for two main reasons:

1. the Gaussian shape requires red only a few parameters to describe it, and

2. the Gaussian shape characterizes the shape of the low-pressure system with sufficient accuracy.

It is clear, of course, that not all possible variations in the cyclone shapes can be described by such simplification. However, we would like to emphasize the importance of other factors in determining sea level variations.

For example, far more relevant to the simulated sea levels than the deviations of the shapes from Gaussian are (1) the depth and extent of the cyclone, and (2) the track and propagation speed of the cyclone.

The cyclone radius of 1000 km is consistent with observations, although we agree that this value is at the upper limit of the distribution. A study by Rudeva and Gulev (2007) found that the effective radius of

cyclones over the ocean can be larger than 900 km, with an average of about 700 km. Thus, we argue that a radius of 1000 km is plausible in the Baltic Sea region. We present this justification in the revised manuscript.

The pressure anomaly ΔP of -40 hPa was chosen so that the resulting maximum surface wind speeds do not exceed 40 m/s. According to geostrophic wind law, larger pressure anomalies would have resulted higher wind speeds, but wind speeds higher than 40 m/s are not plausible in the Baltic Sea region.

In the revised manuscript, we discuss more clearly the limitations of the Gaussian shape and how it differs from a typical extratropical cyclone. As our method is only intended for the Baltic Sea region, where tropical storms cannot occur, we decided not to discuss the similarities with tropical storms or hurricane models.

Rudeva, I., & Gulev, S. K. (2007). Climatology of cyclone size characteristics and their changes during the cyclone life cycle. Monthly Weather Review, 135(7), 2568-2587.

**Numerical Simulations**

The manuscript lacks details about the numerical simulations. It would be helpful to include:

- The computational domain used in the simulations.
- The number of simulations performed, including the range of origins, speeds, and directions

Clarification on the handling of initial conditions

- Currently, the cyclone's origin at 10°E with a 1000 km radius suggests that it affects the Baltic Sea at the start of the computation. The manuscript should explain how initial conditions were handled. Also consider originating the cyclone further away from the Baltic Sea for more accurate results.

The grid size in the simulations

- The current spacing of 0.1 degrees in the meridional direction and 0.2 degrees in the zonal direction may be too coarse for predicting sea level elevations at tide gauge stations.
- The correlation between sea level prediction and grid size should be investigated, considering L159 "The higher spatial resolution Averkiev and Klevannyy (2010) use near St. Petersburg is likely the reason for the 1.5 meter difference between their simulated maximum and our result." If this statement is valid, it means that the sea level elevation may be higher at Oulu and Riga with finer grid simulations.

In the revised manuscript, we have added more information on the details of the numerical simulations, such as the computational domain. The number of simulations varied depending on the location studied. As our aim was not to find the most extreme sea levels, but only to find extremes surpassing the observed ones, the number of simulations is not essential.

We added an explanation on the initial condition, noting that the wind speeds 1000 km away from the cyclone center do not significantly affect the sea levels. Hence, it is not necessary to place the origin of the cyclone further west of the Baltic Sea. We added an explanation of the effect of the grid size, noting

that in an earlier study using the same numerical model, even sea level simulations in sparser grid had correlation over 0.90 with the Finnish tide gauge observations.

It is true that finer grid simulations would likely produce higher extreme sea levels for the sites studied. As our aim was to find lower limits for the maxima, not the most extreme ones, the improved grid simulations are left as a subject for future studies.

**Minor Comments**

- Throughout the text, it seems that the intensity of the cyclone is fixed, but in several places (e.g. L4, L127), "various intensities" are mentioned. The authors need to correct this.

Thank you for pointing this out. We have corrected this in the text.

- Paragraph L50-54 seems unnecessary in the introduction and can be removed.

We removed this paragraph.

- Terms such as "large," "small," and "short" need to be defined more precisely, particularly in L194-197.

The text has been rewritten, L194-197 are in Section 2.2 in the revised version.

- L125 - "As we calculate only short-term sea level changes, our results represent the sea level fluctuation with respect to the sea level preceding the arrival of the cyclone.": unclear

We rewrote this sentence.

- The mention of an observation in 1984 at Oulu in L144 should include an explanation of what caused that anomaly. For example, what was the intensity (pressure and radius) of the related storm?

We added text explaining the physical background of the high sea level in 1984.

**Figures**

- Figure 2 should include labeled x and y axes.

Axis labeling was added

- Figure 3 does not provide valuable information

We agree. Figure 3 was removed

Tables:

- Table 1: Pressure anomaly and radius are fixed for all simulations, thus they are not necessary.

This is true. The table was updated

---

## Author Comment (AC2)

We thank the reviewer for constructive comments of our submitted manuscript. The point-by-point replies to the comments of the reviewer are below. Your comments are marked in black and our responses in red.

Review

On the manuscript "Simulating sea level extremes from synthetic low-pressure systems" by

Jani Särkkä, Jani Räihä, Mika Rantanen, and Matti Kämäräinen.

Overview: Conditional Acceptance Upon Minor Revision

The paper presents simulated conditions of both Baltic sea level hydrodynamics and cyclone pressure conditions in an attempt to better understand extreme sea level variations caused by the latter. The employed method is sound and leads to interesting results. However, the manuscript lacks a proper review of the literature on the methods of extreme value theory on sea level or storm surge extremes as well as in the quantification of these extremes based on the aforementioned theory. Specific comments include:

1. The authors must remind the reader that there already exists a theoretical framework to estimate how much the extremes will deviate from the maximum of a typical Gaussian population. Following the seminal work of Gumbel (1958), the extreme value theory has found important applications on both global and local sea extremes. In global terms, I refer to studies such as of Butler et al. (2007), Arns et al. (2013) and O'Grady et al. (2022).

We recognize the importance of extreme values in the assessment of sea level hazards. The aim of our study was to describe the physical mechanisms leading to extreme sea levels, not to investigate sea levels using extreme value theory. We feel that adding extreme value theory here is beyond the scope of this study but is an important topic in future work. We added discussion explaining the aims and methods of this study at the end of the Introduction.

2A. While the authors correctly point out to the relevance of metocean parameters for the local effect of sea level change, such as wind and currents. To clarify the local effects of wind/waves/currents, the authors should mention that as waves become more nonlinear towards the shore they can decrease the sea level in about 5% of the significant wave height and through wave dissipation increase the sea level in up to 20% of the significant wave height (Bowen et al., 1968; Massel and Gourlay, 2000). For the effect of currents on the mean sea level, the authors could cite at least the theoretical work of Brevik (1978). Additionally, attention has to be made to the fact that extreme waves can further increase this oscillation in mean water level, and estimates from extreme value theory indicate that extreme heights can be increased by 10-30% depending on the sea conditions such as shoaling effects (Benetazzo et al., 2015; Barbariol et al., 2015, 2019; Bolles et al., 2019; Mendes and Scotti, 2020; Trulsen et al., 2020; Mendes et al., 2021), which can further amplify oscillations in the mean water level.

2B. In particular, given a distribution of a time series (let it be the mean sea level for example), one can compute the expected maximum extreme value following section 4b of Benetazzo et al. (2015) or section 3 of Mendes and Scotti (2020). I encourage the authors to attempt to compute this expected maximum of a Gaussian sea for the sea level and compare with their simulations. The authors should

discuss the magnitude of the local wind-wave effects with that of the purely atmospheric pressure. At the very least, I expect the authors to discuss this alternative method.

Thank you for the comments 2A and 2B. The aim of our study was to find lower limits for sea level maxima that go beyond the observed extremes in the Baltic Sea. Therefore, we have not included waves in this study. For more accurate studies for individual locations (as in Apukhtin et al. 2017 and Gordeeva and Klevannyy 2020), wave simulations must be performed separately using the wind fields and simulated sea levels as input data. These studies are left for a future study. We added text explaining why wave studies were not included in this study in the Conclusions.

3. There should be a review of mathematical modelling on cyclone pressure fields, and a discussion of why the particular choice (eq. 1) has been chosen.

The Gaussian shape in eq. 1 was originally chosen for two main reasons:

1. the Gaussian shape requires only a few parameters to describe it, and

2. the Gaussian shape characterizes the shape of the low-pressure system with sufficient accuracy.

In the revised manuscript, we highlight the rationale more clearly and also provide a discussion of other choices made for synthetic cyclones in the literature.

4. The authors provide an ERA5 analysis of cyclone speeds that pass through the Baltic sea, but this information is not sufficient. They also need to display the average spatial (and vector) shape of these cyclones, not to mention the duration of their path in the Baltic Sea. Furthermore, a brief discussion (or figure display) of the intensity of the cyclone as they enter the Baltic Sea.

Thank you for this comment. In the revised manuscript, we have expanded Figure 4 (Fig. 3 in revised version) to include three panels showing the propagation speed, intensity, and duration of cyclones over the Baltic Sea.

The average shape of cyclones in the Baltic Sea would also be an interesting research topic but obtaining the structure of cyclones from e.g. the ERA5 reanalysis would clearly require more methodological resources. In addition, such a study has recently been carried out (Laurila et al., 2021). Therefore, we decided not to calculate cyclone composites for this study.

Laurila, T. K., Gregow, H., Cornér, J., & Sinclair, V. A. (2021). Characteristics of extratropical cyclones and precursors to windstorms in northern Europe. Weather and Climate Dynamics, 2(4), 1111-1130.

5. In section 2.3 the governing equations should be written down, and assumptions and limitations discussed thereof.

The governing equations are the well-known shallow water equations, suitable for the shallow Baltic Sea. We have added explanation for the equations used in the simulations.

6A. The main results are presented in figures 5-6. The text should be clear on whether this analysis can be made only at a few locations, or if these locations were picked for a particular physical reason.

Otherwise, I encourage the authors to provide a Baltic Sea analysis (as claimed in the text) instead of a few locations.

The analyses presented for three sites could be done for any other coastal site, but we chose those sites to represent different bay areas of the Baltic Sea. We modified the text in the beginning of Section 2.5 to clarify this.

6B. While figure 6 is clear on the extremes, it should be normalized by the expected significant wave height as the cyclone passes. This comparison provides a better scale of the cyclone effects as compared to local wind/wave effects on sea level change.

6C. I encourage the authors to provide several contour plot panels showing the scale of normalized sea level change (by the significant wave height) across the entire Baltic sea coast. Each panel would show the sea level change at a particular time since the appearance of the cyclone.

Thank you for comments 6B and 6C. As explained in the earlier comment, the waves are not considered in this study as their effect varies greatly with location and our aim was to find lower limits for the maxima of sea level. For more refined estimates the effects of waves need to be included, this will be a subject of a future study.

6D. Pages 135-145 describe the path of the cyclone affecting the three cities of figure 5. Why not plot

the path on a figure? It would better suit the manuscript and help the reader.

We are confused by this comment. The simulated cyclones have a constant propagation speed, with the origin moving along the straight path indicated by the arrows. So the paths of the three cyclones are described by the three arrows.

Conclusion

The reviewer thanks for the opportunity to read this important work. Overall, I support the publication of this preprint once all these minor issues have been clarified/amended.

Thank you for this positive feedback.

---

## Author Response (AR2)

**Response to reviewer of "Simulating sea level extremes from synthetic low-pressure systems"**

Jani Särkkä, Jani Räihä, Mika Rantanen, and Matti Kämäräinen

We are glad that our major revision has largely satisfied the reviewer. We thank the reviewer for the additional comments and hope that our responses will clarify the rest of the unclear matters.

The point-by-point replies to the comments are below. The comments are marked in black and our responses in blue.

Review on "Simulating sea level extremes from synthetic low-pressure systems" by Jani Särkkä, Jani Räihä, Mika Rantanen, and Matti Kämäräinen

I recommend a revision for this manuscript to be accepted as a publication. My main concerns are on the introduction and result sections.

Major Comments:

Introduction:

The introduction lacks clarity and flow. The author is recommended to restructure the content into clear paragraphs for improved readability and flow. Introduce transition words between sentences to enhance coherence.

The author should consider adding an overview and summarizing the main idea of the manuscript at the end of the introduction.

Thank you for this comment. We have fully revised the introduction of the manuscript. We hope that it is now more readable. The main idea of the manuscript is summarized in the last paragraph of the introduction, as suggested by the reviewer.

In L42-51, the arguments are challenging to follow, and the meaning of "statistical methods" is unclear. To improve, the author should clarify the concept of statistical methods and explain how they relate to natural hazards, particularly focusing on whether they consider outliers or medium-sized events for estimating extreme sea levels.

The phrase "to estimate the lower limits of extreme sea levels" needs clarification for better understanding.

As indicated in the previous answer, we rewrote the introduction text to clarify our objectives. We replaced "statistical methods" with "extreme value analysis methods" and removed the phrase referring to lower limits of extreme sea levels.

At L121, the statement regarding the Baltic Sea and "shallow water" is incorrect. The term "shallow water" involves considerations beyond depth, such as wavelength and amplitude compared to depth. Either remove the sentence or strengthen the argument by addressing these factors.

We agree. We removed this sentence.

Results:

Move the content in L163-165 to the previous section and modify it to mention the limitations of the current study.

We moved the content to Section 2.5 and modified the text (L156-L160)

In L187 and L195, replace "originate" with more appropriate terms like "progress," "propagate," or "approach".

We changed the wording using verbs "enter" and "approach"

In L191, if comparing numerical results, provide details of the study by Averkiev and Klevannyy (2010), including storm speed, pressure, wind speed, etc.

Thank you for this comment. We added details on L191-199 of the numerical parameters used in Averkiev and Klevannyy (2010) to make comparison with our simulation methods.

In L191, if the authors claim that the grid size is the primary reason for the difference, then the authors need to provide more plausible explanations why the grid size is important in St. Petersburg, and not important at other locations.

As we were not sure about the reason, we removed the sentence referring to the spatial resolution in the Averkiev and Klevannyy (2010) study.

Minor Comments:

In L21, replace "it may be assumed" with "it is possible" for clarity and precision.

We modified the text according to the comment.

At L34, it would be beneficial for the authors to provide the average annual number of occurrences instead of just stating "The annual number of."

We modified the text and mention now that there are about five windstorms per month in northern Europe (L34-35)

Remove the term "Somewhat subjectively" at L103, and instead, support statements with references.

Clarify the reason behind "not plausible in the Baltic Sea region" at L106 by providing a reference.

We rewrote the text (L104-108) justifying the limit of 40m/s for mean wind speed and added references.

At L110, explicitly state that the authors fix the radius, wind speed, and the magnitude of the central pressure.

We do not fix directly the wind speed, it is limited by the fixed pressure anomaly and cyclone radius. We added in the text mention on the fixed parameters.

Change "This sea level model" to "which" at L119.

Suggested change would have resulted in a long sentence; we modified the text to increase fluency.

Explain the meaning of "correlation" at L125.

We added explanation in the text.

L141 seems unclear; consider rewriting or removing it.

Remove "entire" from L142.

We rewrote the text to make it clearer.

Replace "largest" with "large" at L146.

We made the suggested replacement.

L161 "For even lower … be still higher" Provide the evidence for this statement. assumptions, the vertical velocity can be ignored. Remove that sentence or strengthen the argument.

We removed the sentence.